# Study of Motion Sickness Model Based on fNIRS Multiband Features during Car Rides

**DOI:** 10.3390/diagnostics13081462

**Published:** 2023-04-18

**Authors:** Bin Ren, Wanli Guan, Qinyu Zhou

**Affiliations:** Shanghai Key Laboratory of Intelligent Manufacturing and Robotics, School of Mechatronic Engineering and Automation, Shanghai University, Shanghai 200444, China; milessg@shu.edu.cn (W.G.); zqy1101@shu.edu.cn (Q.Z.)

**Keywords:** motion sickness, functional near-infrared spectroscopy (fNIRS), principal component analysis (PCA), wavelet decomposition, power spectral entropy (PSE), support vector machine (SVM), motion sickness model

## Abstract

Motion sickness is a common physiological discomfort phenomenon during car rides. In this paper, the functional near-infrared spectroscopy (fNIRS) technique was used in real-world vehicle testing. The fNIRS technique was utilized to model the relationship between changes in blood oxygenation levels in the prefrontal cortex of passengers and motion sickness symptoms under different motion conditions. To enhance the accuracy of motion sickness classification, the study utilized principal component analysis (PCA) to extract the most significant features from the test data. Wavelet decomposition was used to extract the power spectrum entropy (PSE) features of five frequency bands highly related to motion sickness. The correlation between motion sickness and cerebral blood oxygen levels was modeled by a 6-point scale calibration for the subjective evaluation of the degree of passenger motion sickness. A support vector machine (SVM) was used to build a motion sickness classification model, achieving an accuracy of 87.3% with the 78 sets of data. However, individual analysis of the 13 subjects showed a varying range of accuracy from 50% to 100%, suggesting the presence of individual differences in the relationship between cerebral blood oxygen levels and motion sickness symptoms. Thus, the results demonstrated that the magnitude of motion sickness during the ride was closely related to the change in the PSE of the five frequency bands of cerebral prefrontal blood oxygen, but further studies are needed to investigate individual variability.

## 1. Introduction

Motion sickness is a common symptom that includes dizziness, fatigue, loss of appetite, gastrointestinal distress, and vomiting [1]. Its duration can vary from a few minutes to several hours [2]. Although motion sickness usually does not lead to serious harm, severe cases can cause discomfort, including headaches and cold sweats, which may negatively impact one’s health [3]. The direct causation of motion sickness is thought to come from the conflict between human expectations and measurements of spatial orientation rather than a specific sensory modality [4]. The prefrontal cortex (PFC) plays an essential role in the level of driving, as indicated by differences in brain activity observed in fNIRS studies [5]. In addition, the altered blood supply to the local area is an important direct manifestation of neuronal dysfunction [6]. Therefore, the study of blood supply to the prefrontal region of the brain is of great importance.

Since the 1970s, researchers have been studying motion sickness by analyzing physiological signals with methods such as the electroencephalogram (EEG), the electrocardiogram (ECG), heart rate variability (HRV), and the electrogastrogram (EGG) [7]. Zhao, L. et al. investigated the correlation between automatic and active driving by collecting motion sickness scores and EEG signals from subjects during automatic and active driving. The results showed that subjects had higher motion sickness scores on average under automatic driving than under active driving. As the degree of motion sickness increased, the frequency of the center of gravity of the brain motor center (FC2, Cz) theta wave power spectral density increased [8]. Lim, H.K. et al. evaluated the reliability of using the EEG to evaluate motion sickness and identified specific waves and regions [9]. Li, Z. et al. further demonstrated the validation of the mean EEG frequency of theta band as an indicator of motion sickness [10]. Ji-Un, H. et al. fused five physiological signals—EEG, ECG, respiration (RESP), photoplethysmography (PPG), and galvanic skin response (GSR)—to construct a motion sickness classification model based on real-world vehicle testing [11]. Although the use of various physiological signals is a more objective response to the level of motion sickness, it has been suggested that HRV and GSR may be controversial indexes that need to be reassessed before they can be used again for the assessment of motion sickness [12]. In addition, Tan, R. et al. investigated motion sickness detection using physiological signals in on-road driving scenarios [13].

Motion sickness is thought to be caused by a conflict between the vestibular, visual, and other proprioceptive systems, which are involved in the perception of motion [14]. The prefrontal area of the brain is important to human perception [15]. Therefore, the fNIRS technique was adopted in this study.

fNIRS is a noninvasive, real-time technique for monitoring metabolic activity in living organisms [16]. It can measure blood oxygenation and related parameters in biological tissues [17]. The fNIRS technology detects changes in the concentration of oxyhemoglobin (HbO) and deoxyhemoglobin (HbR) in tissues by measuring the absorption of near-infrared light in the near-infrared wavelength range, thus enabling the monitoring of the functional activity of biological tissues [18]. In recent years, the fNIRS technique has been widely used in medicine, exercise physiology, neuroscience, nutrition, and bioengineering [19]. Compared with other traditional physiological monitoring techniques, fNIRS has advantages such as high spatial resolution ranging from 1 to 3 centimeters, temporal resolution ranging from 100 milliseconds to 1 s, and ease of operation [20]. This study aims to investigate the potential of fNIRS technology for the study of motion sickness with the goal of improving our understanding, diagnosis, and treatment of this condition. 

Research on motion sickness and cerebral blood oxygen concentration has made some progress nationally and internationally. Zhang et al. (2019) showed that the Brodmann area of the brain associated with motion sickness reacts differently under straightforward working conditions. The findings contribute to the study of motion sickness from a brain perspective [7]. Zhang’s team again designed environments for straight and curved driving conditions in 2020, performed driving maneuvers, and recorded driver brain activity to investigate the mechanisms correlating motion sickness with cortical activity. The findings suggest that the emergence of the motion sickness response is associated with the occipital lobe and propose a new research method to investigate the correlation mechanism between motion sickness and cortical activity [7]. The study by Golding, J.F. et al. has increased knowledge of the environmental and behavioral conditions that trigger and prevent motion sickness as well as the sensitivity of vestibular disorders to motion sickness and the associated brain mechanisms [21]. Gavgani, A.M. et al. reported a significant increase in HbO concentrations in channel 2/15 in the parieto-temporal regions of both hemispheres in subjects presenting with motion sickness. Increased HbO concentrations were associated with increased nausea and motion sickness symptoms. The study suggested that motion sickness leads to complex changes in cerebral blood flow, with increased perfusion in some cortical regions but decreased overall cerebral perfusion [6]. 

The purpose of this study is to establish a model that examines the relationship between motion sickness and prefrontal cerebral blood oxygen concentration. This is accomplished by using fNIRS technology to measure blood oxygen concentration in the prefrontal cortex as well as by collecting data on motion sickness symptoms during real-world vehicle testing supported by SAIC Group.

## 2. Materials and Methods

### 2.1. System Architecture Diagram

The architecture of the entire system is shown in Figure 1. This study aimed to develop a classification model for the degree of motion sickness. The entire process consists of 4 main steps. First, the cerebral blood oxygen signal was preprocessed, including mean filtering, median filtering, and motion artifact suppression (MAS), to obtain the preprocessed data. Second, one of the most significant feature channels was obtained by PCA. Third, wavelet decomposition was performed on the channel to obtain the data of frequency bands I–V. The features of PSE are extracted based on different frequency bands. Fourth, the data were fed into SVM for machine learning by combining a 6-point subjective evaluation of the motion sickness scale. Through validation, a motion sickness model was obtained to achieve the classification of the degree of motion sickness.

### 2.2. fNIRS Signal and Preprocessing

fNIRS takes advantage of the better transmittance of NIR light in tissues by placing multiple light sources and light detectors on the scalp to record the intensity changes of NIR light as it passes through the scalp, skull, and brain tissue. The operating principle of fNIRS is based on two different near-infrared light absorption spectra, one at around 760 nm, absorbed by HbO and HbR, and the other at around 850 nm, absorbed only by HbO. Figure 2 shows a subject wearing the fNIRS device, and the magnified portion on the right side of Figure 2 displays the measurement principle of the fNIRS device. Specifically, the red section on the left represents the “light source,” which includes two types of near-infrared light in the 730 nm and 850 nm wavelength bands, passing through the brain tissue of the subject’s forehead and finally being received by the light-intensity sensor, or “Detector”. By simultaneously measuring the intensity changes of NIR light at these 2 wavelengths and using the Lambert–Beer law formula [19], changes in the concentrations of HbO and HbR can be calculated, thus reflecting changes in the level of metabolic activity in brain regions.
(1)$$\frac{\Delta I_{1}}{I_{1}}-\frac{\Delta I_{2}}{I_{2}}=\epsilon_{HbR}\Delta C_{HbR}-\epsilon_{HbO}\Delta C_{HbO}$$
where $∆C_{HbO}$ and $∆C_{HbR}$ represent changes in the concentrations of HbO and HbR, respectively. $\epsilon_{HbO}$ and $\epsilon_{HbR}$ denote the molar absorption coefficients of the HbO and HbR, respectively, and the variables $I_{1}$ and $I_{2}$ denote the variation of the NIR light intensity at 2 wavelengths. Based on Equation (1) above [22,23], the amount of change in the concentration of HbO and HbR, and consequently the concentration of oxyhemoglobin, can be calculated by measuring the change in NIR light intensity at 2 wavelengths, $∆I_{1}$ and $∆I_{2}$.

The first 5 min of data were excluded from this study. A series of pretreatment steps were used in this study. First, the raw light-intensity signal of each channel was mean filtered (MeanFilt) and median filtered (MedFilt) to reduce the effect of high- and low-frequency noise. The main frequency components of the signal were also retained to better reflect the actual variation of the physiological signal.

In addition, when subjects are performing fNIRS acquisitions, movements of their head or body cause relative changes in the distance between the optical probe and the area being measured. These changes lead to motion artifacts in the fNIRS signal. To solve this problem, this study uses discrete wavelet transform to perform motion artifact suppression [24] on the 6-channel fNIRS signal, thresholds the wavelet coefficients obtained after several decompositions, and takes out the wavelet coefficients that exceed the thresholds and reconstructs them to achieve the removal of motion artifacts. This method can effectively improve the quality and reliability of the signal and provide a more accurate basis for subsequent data analysis [24]. As shown in Figure 3a–d, the fNIRS data preprocessing procedure included MeanFilt, MedFilt, and MAS to obtain the preprocessed data, which was then used for subsequent analyses. In addition, to more clearly observe the effects of data preprocessing, Figure 3e shows the light intensity signal diagrams of MeanFilt, MedFilt, and MAS in the same figure sequentially.

### 2.3. Channel Dimensionality Reduction Based on PCA

The signal-to-noise ratio of HbO is higher than that of HbR, which is more sensitive. In addition, when the two signals are combined for analysis, this does not improve the accuracy of the results much. Thus, this study was conducted to construct a motion sickness model by analyzing the changes of HbO content in the prefrontal area of the brain [25]. The fNIRS device has a total of 6 channels. The relative amount of change in HbO can be calculated by acquiring the change in absorbed light intensity based on the Lambert–Beer law algorithm.

The fNIR103E device has 6 sampling channels, and the six channels are correlated with each other and have some redundancy. Therefore, this study uses PCA [26] to downscale the data of the original 6 channels to obtain the single-channel information with the most significant features [22].

PCA is a common data dimensionality reduction technique. Its basic idea is to project the original data onto a new set of coordinate systems, which can be obtained as PC1 and PC2, the first and second principal components obtained in PCA, respectively. PC1 and PC2 are the new variables obtained from the original variables after linear transformation. PC1 is the direction of maximum variance obtained by linear combination of the original variables, that is, the direction of maximum variance of the data. PC2 is the direction orthogonal to PC1 and is the direction of maximum variance of the residuals in the direction of PC1, that is, the linear combination of maximum variance outside the first principal component.

In this study, the scikit-learn library in Python was used for PCA analysis, and the number of retained principal components was selected based on the cumulative variance contribution and Kaiser criterion. Usually, the first two principal components will explain most of the variation of the data. Therefore, the channel corresponding to the maximum value can be taken as the principal component channel.

As shown in Figure 4, the projection values for each channel on PC1 and PC2 are calculated, with PC1 and PC2 being the first and second principal components obtained from Subject 1’s 6 samples. This study used PCA to downscale the original 6-channel data to obtain the single-channel information with the most significant features. By normalizing the projection values on PC1 and PC2, the relative size and orientation of each channel were determined. Both PC1 and PC2 have values ranging from −0.6 to 0.68, and the most significant channels in each segment are shown as red circles in Figure 4.

### 2.4. Feature Extraction of Multiband PSE

For cerebral blood oxygen signals, different frequency bands originate from different physiological activities, and each frequency band has a specific physiological meaning [27,28]. Currently, the cerebral blood oxygen signal is a time series signal in the main range of 0–2 Hz. These signals can be divided into 5 frequency bands and used for the study of motion sickness classification. The specific frequency bands are divided as shown in Table 1. These frequency bands have an important role in the study of motion sickness [29,30].

According to Nyquist’s sampling theorem, the sampling frequency should be greater than 2 times the highest frequency of signal extraction, that is, fs.max>2fmax [31]. Only in this case can the sampled signal retain the information in the original signal intact; otherwise, sampling distortion will occur [32].

Since the cerebral blood oxygen signal collected by the NIR experimental equipment is in the time domain, converting the time domain signal to the frequency domain signal is the best way to gain insight into the relationship between blood oxygen changes in the prefrontal area and the degree of motion sickness [33,34]. In this study, the single channel basis with the most significant correlation features screened by PCA in the previous section was selected, and wavelet decomposition was performed on this channel. The cerebral blood oxygenation signal is a nonstationary signal, which cannot be analyzed by the conventional Fourier Transform [35]. The Short Time Fourier Transform [36], on the other hand, has a single resolution due to its fixed window function. To clarify, the "infinite-length trigonometric function" in the Short Time Fourier Transform can be replaced with a "finite decayable wavelet basis" to achieve time-frequency conversion. This enables the signal features to be adjusted to the appropriate window size, resulting in enhanced resolution.

Wavelet transform has multiscale and multiresolution features, which can gradually focus and amplify the signal to be analyzed and can distinguish local features [36]. The signal f(t) is represented or approximated by a certain function, which is composed of the basic wavelet basis function ψ(t) scaled and translated by different scales. Let $\psi (t)\epsilon L^{2}(R)$, when ψ(ω) satisfies the allowable condition [37], be: (2)$$C\psi =\int \frac{\left|\psi (\omega )\right|}{\left|\omega \right|}<\infty$$

Then, *ψ*(*t*) can be called a fundamental wavelet or mother wavelet function. The wavelet sequence is obtained by stretching and translating it [37]:(3)$$\psi_{a0b}\left(t\right)=\frac{1}{\sqrt{\left|a\right|}}\psi (\frac{t-b}{a})$$

In the equation, a and b are the scaling parameter and the scale parameter, respectively. Since the cerebral blood oxygen signal consists of discrete sampling points, a and b are then discretized. Take $a={a_{0}}^{m},b=nb_{0}{a_{0}}^{m}$, where m and n are integers, $a_{0}>1$ is a fixed scaling step, $b_{0}>0$, and is related to the specific form of the mother wavelet. Thus, the discrete wavelet function can be defined as [37]:(4)$$\psi_{m0n}\left(t\right)={a_{0}}^{-\frac{m}{2}}\psi \left(\frac{t-nb_{0}{a_{0}}^{m}}{{a_{0}}^{m}}\right)={a_{0}}^{-\frac{m}{2}}\psi ({a_{0}}^{-m}t-nb_{0})$$

The wavelet basis function used in this study is Daubechies wavelet because of its excellent properties in orthogonality, time-frequency tight support orthogonality, etc. To obtain a better signal decomposition effect, db5 with 5th-order vanishing moment in Daubechies wavelet series was chosen as the wavelet basis function in this study. In order to determine the number of layers of decomposition, it is necessary to combine the sampling frequency and the range of frequency bands. According to the previously mentioned range of frequency bands I–V, the PSE features of 5 frequency bands need to be extracted in this study, and the sampling frequency of the fNIRS equipment is 10 Hz, so 9 layers of wavelet decomposition are selected. Figure 5 shows the number of wavelet decomposition layers and the corresponding frequency band ranges.

As shown in Figure 5, after wavelet decomposition, the timing diagram corresponding to the frequency bands I–V can be obtained, and the 9-layer wavelet decomposition are obtained as approximation coefficients (a1–a9) and detail coefficients (d1–d9). The detail coefficients, which contain the high-frequency part of the original signal, can provide information about signal variations and local features. Wavelet decomposition of the cerebral blood oxygen signal yields the detail coefficients d for different frequency bands, where the value of detail coefficient d3 corresponds to frequency band I (0.6–2.0 Hz), the value of d5 corresponds to frequency band II (0.145–0.6 Hz), the value of d6 corresponds to frequency band III (0.052–0.145 Hz), the value of d8 corresponds to frequency band IV (0.021–0.052 Hz), and the value of d9 corresponds to band V (0.0095–0.021 Hz). By analyzing the time domain diagram of the 5-band signal, an in-depth understanding of the characteristics and various patterns of the signal can be obtained. Effective features can also be extracted. The relationship between signals in different frequency bands and physiological activities can be further explored to provide a theoretical basis and reference for related research.

So far, the wavelet transform method has obtained the cerebral blood oxygen timing maps of 5 frequency bands. Next, we select the PSE of the 5 frequency bands as the features of the motion sickness classification model. PSE is a method for calculating the spectral features, and by calculating the spectral density of cerebral blood oxygen signals, we can obtain information such as the complexity and equilibrium of cerebral blood oxygen signals. In this study, PSE was selected as the feature basis because it has better robustness and reliability [38], can better reflect the characteristics and differences of signals in different frequency bands [39,40], can be used as a new nonlinear measure of signal complexity, and can be used for feature extraction and classification of signals [40] to provide effective feature information for the establishment of motion sickness classification models.

For time domain discrete sequence $x\left(n\right),n=0,1,\ldots ,N-1$, the PSE is denoted as [41]:(5)$$S_{x}=\frac{1}{N}\left|{x(k)}^{2}\right|,k=0,1,2,\ldots ,N-1$$

The energy is conserved during the transformation of the signal from the time domain to the frequency domain, that is [41],
(6)$$\sum_{n=0}^{N-1}{\left|x_{n}\right|}^{2}=\sum_{k=0}^{N-1}{\left|S_{x}(k)\right|}^{2}$$

Considering $S_{x}\left(k\right),k=1,2,\ldots ,3,N$ as an energy division of the signal $x\left(t\right)$ in the instantaneous frequency domain space, the PSE of the signal can be calculated as [41]:(7)$$F_{SE}=-\sum_{k=1}^{N}p_{k}{log}_{2}p_{k}$$
(8)$$p_{k}=S_{x}(k)/\sum_{k=1}^{N}S_{x}(k)$$
where $p_{k}$ denotes the percentage of the kth power spectrum in the whole power spectrum and also denotes the frequency of the kth frequency corresponding to the amplitude occurrence. The normalized PSE is [41]:(9)$$F_{PSE}=\frac{F_{SE}}{{log}_{2}N}$$

The above Equations (5)–(9) can be derived from the PSE corresponding to the Ⅰ–V frequency band. So far, the feature extraction of the PSE of multiple frequency bands has been completed, providing the feature input of data for the following construction of the motion sickness classification model, realizing the automatic identification and evaluation of the degree of passenger motion sickness.

### 2.5. Motion Sickness Model Construction Combined with Subjective Evaluation

In this study, subjects were counted every 5 min on a 6-point scale of subjective measures of motion sickness [42] in real-world vehicle testing. The 6-point motion sickness scale, as shown in Table 2, includes “no motion sickness at all,” “slight motion sickness,” “some motion sickness,” “much motion sickness,” “extreme motion sickness,” and “unbearable motion sickness.” Each degree was scored separately, corresponding to a scale of 0–5. For statistical purposes, a score of 0–1 is considered as no motion sickness (labeled as 0 in SVM), and a score of 2–5 is considered as motion sickness (labeled as 1 in SVM) [43]. Therefore, this study belongs to a typical dichotomous model for motion sickness discrimination.

SVM [44] is a commonly used machine learning algorithm for binary and multiclassification tasks. The basic principle is to find an optimal hyperplane (decision boundary) in a high-dimensional space to separate different classes of data [45]. Specifically, for the binary classification problem, the SVM tries to find an optimal hyperplane such that all data points belonging to Category 1 lie on one side of the hyperplane and all data points belonging to Category 2 lie on the other side.

The input feature of this study is the PSE of frequency bands I-V, and the output label is the degree of motion sickness of passengers. Specifically, the multiband PSE feature vector of each subject is used as a training sample, and its corresponding motion sickness degree is used as a label.

The decision function of SVM can be expressed as [46]:(10)$$f\left(x\right)=sign(w^{\wedge }Tx+b)$$
where x is the feature vector of the input samples, w is a weight vector, and b is a bias term. The objective of the SVM is to maximize the interval between the hyperplane and the nearest sample point, that is, to minimize the norm of w, while making all samples satisfy the classification requirements. Specifically, the SVM aims to make the objective function minimized [46]:(11)$${min}_{\omega }\frac{1}{2}{\left|\left|\omega \right|\right|}_{2}^{2}\xi_{i}+C\sum_{i=1}^{m}\xi_{i}$$
where C is a penalty coefficient to control the balance between the classification error and the decision boundary, and $\xi_{i}$ is a slack variable to deal with non-distinguishable sample points.

SVM makes a linearly indivisible problem linearly divisible in a high-dimensional space by mapping the input samples into the high-dimensional space. Specifically, SVM uses a kernel function to compute the inner product of the input samples in the high-dimensional space to obtain a nonlinear decision boundary.

In this study, the Radial Basis Function (RBF) is used as the kernel function to calculate the inner product of the input samples in the high-dimensional space by using the following equation [46]:(12)$$K\left(x,x^{\prime }\right)=exp({-\gamma \left|\left|x-x^{\prime }\right|\right|}^{\wedge }2)$$
where γ is a parameter that controls the shape of the RBF kernel function.

The SVM algorithm is used to construct a motion sickness binary classification model using the PSE feature vector of multiple frequency bands as the input and the motion sickness degree as the output label.

## 3. Passenger Motion Sickness Experimental Design

The test is aimed to collect data from 13 subjects who had no medical history of relevant diseases and did not use any psycho-stimulating substances such as coffee, strong tea, or drugs within 24 hours before the experiment. The subjects included 6 males and 7 females with an age range of 26 to 32 years.

The experiment was conducted using actual road tests to simulate real-world vehicle riding. The experimental design scheme is shown in Figure 6. During driving, the subjects were required to sit still in their seats for 5 min, followed by a 30-min riding session with a 20-min break in between for the next set of experiments.

In order to reduce the influence of external factors on the experimental results, only 1 subject was arranged for each day, and the experimental environment was chosen as “Outer Ring-S7-Baoan Highway–Hutai Road Loop” in Baoshan District, Shanghai, with a total distance of about 12.1 km. To avoid the relief of motion sickness by frequent traffic light starting and stopping, this study conducted the test between 2:00 p.m. and 4:00 p.m., when traffic light is light. The experimental vehicle chosen was the F7 series of new energy vehicles provided by SAIC. Figure 7 shows the test vehicle, route, and pictures of the left, front, and right views of one subject for the real-world vehicle test experiment. In addition, we also recorded some underlying assumptions during the experiment, such as fixed speed intervals, driving time, and driver experience, which can be provided to readers if needed.

During driving, the lead driver adjusts the power response and regen (regenerative braking) intensity of the vehicle; subjects are not informed about these adjustments. Subjects fill out a 6-point scale for subjective evaluation of motion sickness every 5 min to record the degree of motion sickness at the current time point. Consent was obtained from the subjects before the experiments were conducted, and the ethical requirements for human subjects were strictly observed during the experiments.

The study evaluated the classification model using a 5-fold cross-validation approach by dividing the dataset into 5 parts, selecting 4 of them for training and the remaining 1 part for testing each time and repeating the process 5 times, selecting a different test and training set each time. The confusion matrix can be used to compare the differences between the model predictions and the true results [47]. As shown in Figure 8a–d, the confusion matrix illustrates the classification results of dizziness severity for 13 subjects. The rows of the matrix represent the true classes of the subjects, while the columns represent the predicted classes. The diagonal elements indicate the number of subjects whose predicted dizziness severity matches their actual severity, while the off-diagonal elements indicate the misclassified subjects. It is worth noting that the accuracy rate for the 13 subjects varies, with the highest accuracy being 100% and the lowest being for Subject 2, with only a 50% accuracy rate. In this motion sickness model, the confusion matrix divides the true and predicted outcomes into 4 categories, true positive (T.P.), false positive (F.P.), true negative (T.N.), and false negative (F.N.), respectively. True positive refers to the number of samples where the model correctly predicts positive cases, false positive refers to the number of samples where the model incorrectly predicts negative cases as positive cases, true negative refers to the number of samples where the model correctly predicts negative cases as negative cases, and false negative refers to the number of samples where the model incorrectly predicts positive cases as negative cases. These results can be represented in the form of a matrix. As shown in Figure 9, a confusion matrix is plotted for 78 sets of samples, where 32 samples are correctly classified as negative cases (no motion sickness) and 36 samples are correctly classified as positive cases (motion sickness). Of these, 68 samples were correctly classified, resulting in an accuracy rate of 87.18%. T.P., T.N., F.P., and F.N. denote the number of true, true negative, false positive, and false negative cases, respectively, with a recall metric of 0.9143 and an F1 value of 0.8852.

## 4. Discussion

This study has some limitations that should be acknowledged. First, data collection was performed through real-world road tests, which may be affected by various factors such as driving route, speed, and environment. These factors may lead to variations in the collected data and limit the generalizability of our findings. Future studies could consider using simulators or virtual reality technologies to control for these variables and improve the validity of the data.

Second, our study employed a small sample design, which may limit the statistical power and generalizability of our conclusions. Future research could adopt larger sample sizes and multicenter studies to enhance the reliability and generalizability of our findings.

Finally, while PCA and wavelet decomposition were used to extract features from the prefrontal blood oxygen signals in our study, other more advanced algorithms could be explored for further data analysis, such as deep learning methods and independent component analysis. These advanced algorithms may provide more accurate and reliable results for future studies in this field.

Despite these limitations, our study contributes to the understanding of motion sickness and provides insights for developing effective interventions to reduce its incidence. Future studies could build upon our findings and address these limitations to further advance this field of research.

## 5. Conclusions

This study intended to investigate the relationship between prefrontal blood oxygen signal and motion sickness during riding. It was found that there was a significant correlation between the PSE characteristics of the subjects’ brain prefrontal in five frequency bands and the subjects’ degree of motion sickness. Further information was extracted through PCA and wavelet decomposition. PCA helped this study to extract the most significant features from the six channels, and wavelet analysis helped us to determine the appropriate number of stratifications. The HbO values of different frequency bands were also analyzed. In this study, data were collected from 13 subjects using BIOPAC’s fNIRS cerebral oximeter with real-world road tests. The results of the study suggest that the magnitude of motion sickness is strongly correlated with changes in the entropy of the PSE of the five frequency bands of cerebral prefrontal blood oxygen. To further investigate the relationship between each frequency band and the classification accuracy, we used correlation analysis. The results indicated that there was a certain degree of correlation between the PSE characteristics in frequency bands I, III, and IV and the subjective evaluation of motion sickness, with correlation coefficients of 0.6, 0.2, and 0.19, respectively. This finding could be beneficial for understanding the development of motion sickness and exploring potential methods to alleviate it during car rides.

## Figures and Tables

**Figure 1 diagnostics-13-01462-f001:**
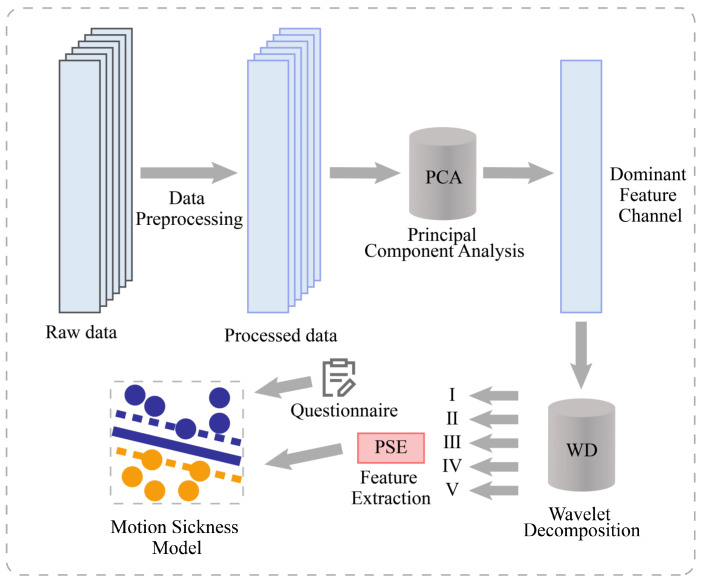
System architecture diagram.

**Figure 2 diagnostics-13-01462-f002:**
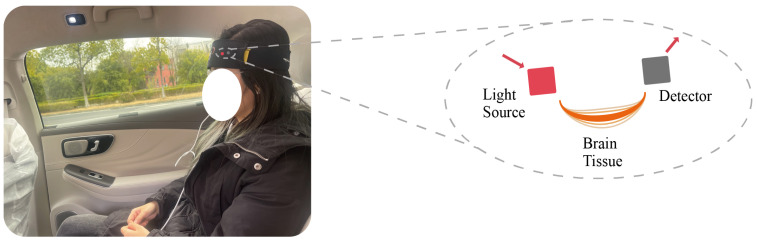
fNIRS device and measurement principle with Subject.

**Figure 3 diagnostics-13-01462-f003:**
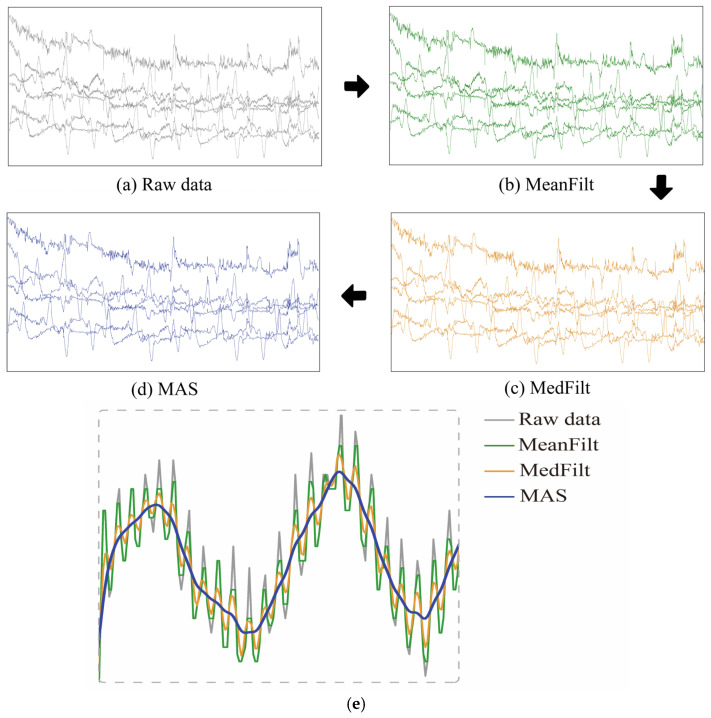
(**a**–**d**) The fNIRS data preprocessing procedure, including the original data, Meanfilt data, MedFilt data, and MAS-corrected data. (**e**) Raw light-intensity signal, MeanFilt, MedFilt, and MAS.

**Figure 4 diagnostics-13-01462-f004:**
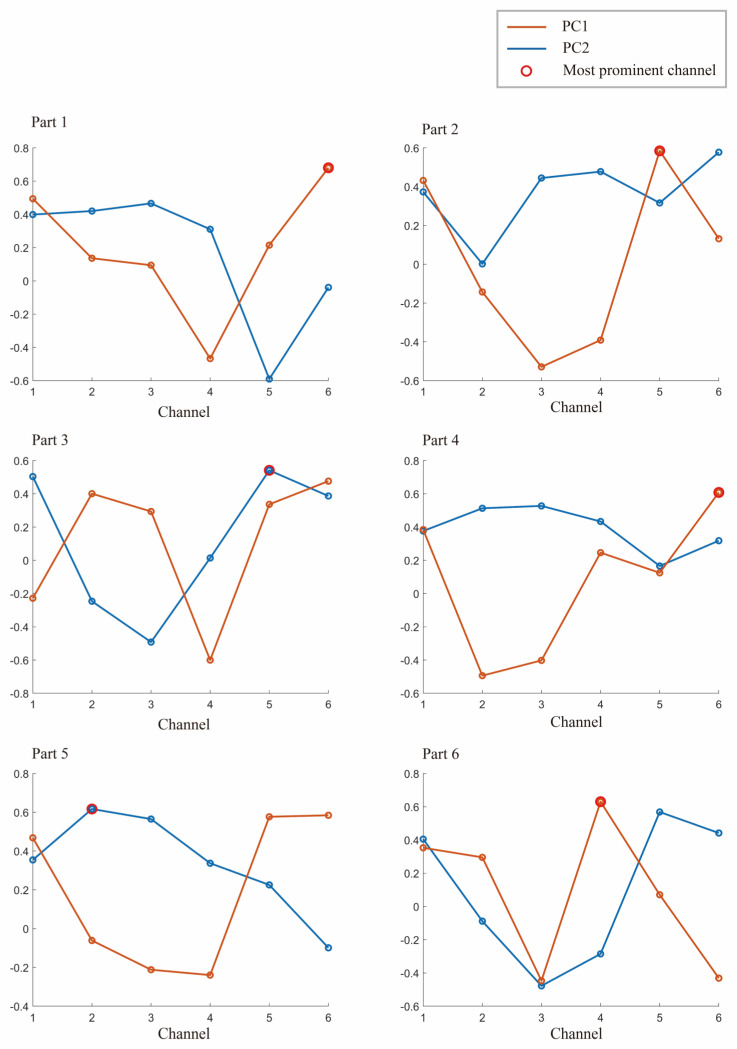
The first component PC1 and the second principal component PC2 of Subject 1 in a set of experiments. The largest of them is marked with a red circle, and the channel corresponding to the red circle is selected for the specified channel.

**Figure 5 diagnostics-13-01462-f005:**
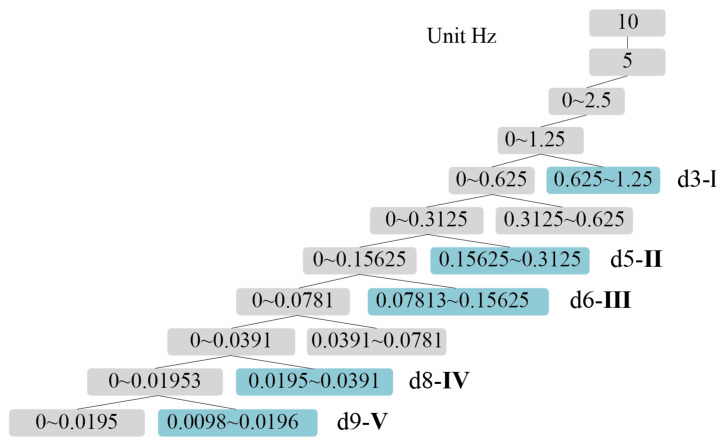
Relationship between wavelet layers and I–V band range.

**Figure 6 diagnostics-13-01462-f006:**
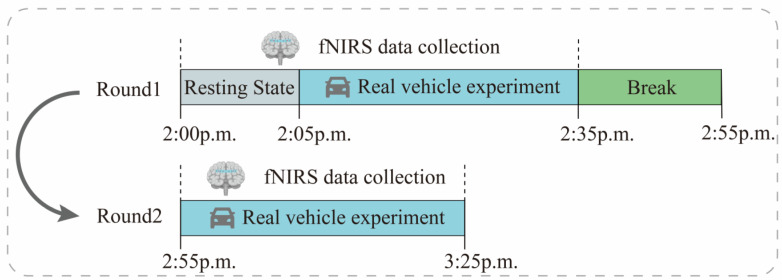
Experiment flow chart with fNIRS signal recording.

**Figure 7 diagnostics-13-01462-f007:**
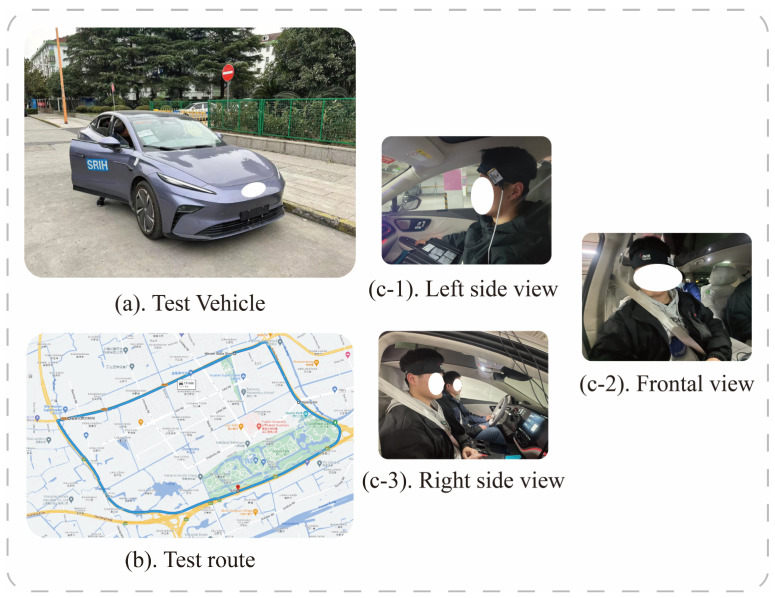
(**a**) is the test vehicle, (**b**) is the test route, and (**c-1**–**c-3**) denotes one of the subjects.

**Figure 8 diagnostics-13-01462-f008:**
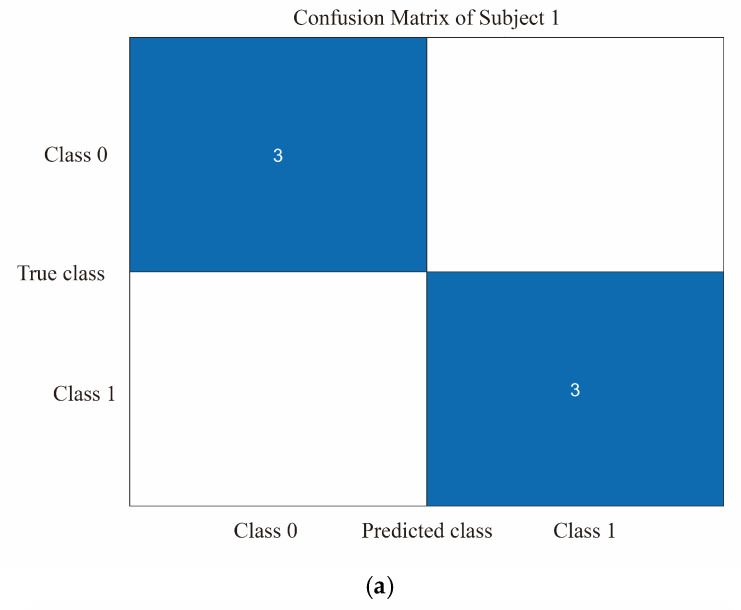
(**a**) Classification of dizziness severity for Subject 1. (**b**) Classification of dizziness severity for Subjects 2–5. (**c**) Classification of dizziness severity for Subjects 6–9. (**d**) Classification of dizziness severity for Subjects 10–13.

**Figure 9 diagnostics-13-01462-f009:**
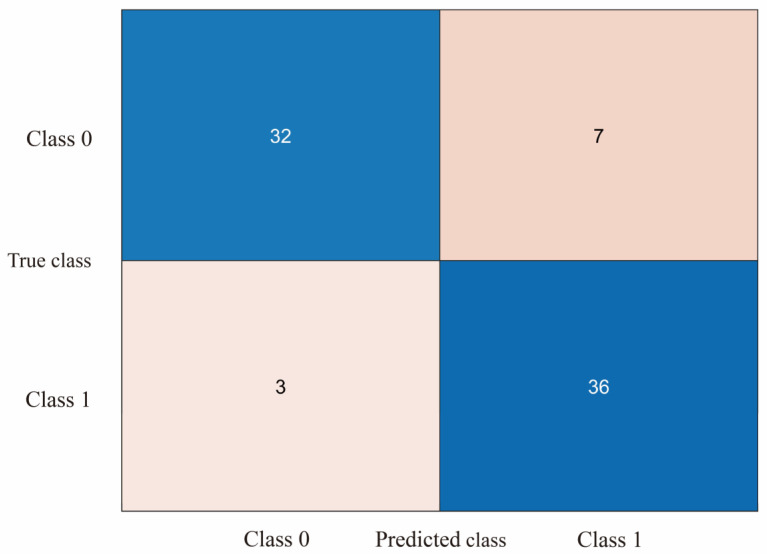
Confusion matrix plot of SVM output for all subjects’ motion sickness levels.

**Table 1 diagnostics-13-01462-t001:** Band range of physiological signals in the five frequency bands.

Frequency Band	Frequency (Hz)	Physiological Meaning
I	0.6–2.0	Heart rate activity
II	0.145–0.6	Respiratory activity
III	0.052–0.145	Myogenic activity
IV	0.021–0.052	Neurogenic activity
V	0.0095–0.021	Endothelial cell metabolic activity

**Table 2 diagnostics-13-01462-t002:** The -point motion sickness scale: Descriptions and corresponding scores for different degrees of motion sickness severity.

Score	Description
0	No motion sickness at all: participants have no sensation of motion sickness and feel normal.
1	Slight motion sickness: participants have slight motion sickness but can still carry on with normal activities.
2	Some motion sickness: participants experience motion sickness symptoms and may feel some discomfort but can still carry on with normal activities.
3	Much motion sickness: participants experience moderate motion sickness symptoms such as dizziness and nausea, which affect their ability to carry on with normal activities.
4	Extreme motion sickness: participants experience severe motion sickness symptoms such as severe dizziness and nausea, which prevent them from carrying on with normal activities.
5	Unbearable motion sickness: participants experience extremely severe motion sickness symptoms such as unbearable dizziness, nausea, and vomiting, which require immediate cessation of the activity and medical attention.

## Data Availability

All data included in this study is available upon request by contact with the corresponding author. The data is not publicly available because of ethical restrictions.

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
