# Peer review of "Study of Motion Sickness Model Based on fNIRS Multiband Features during Car Rides"

_diagnostics, 2023, doi:10.3390/diagnostics13081462_

Round 1

Reviewer 1 Report

Overall, the study is interesting.

There are accuracy issues regarding citations that the authors show as supporting a statement,

The methodology is also far from clear, particularly with respect to how the data is processed.

The results are not detailed enough. For instance, I would expect at least a plot of sickness with respect to time with the classification on the validation set

More detailed feedback is provided in the annotated pdf:

Author Response

Dear Reviewer,

    First and foremost, we would like to express our sincere gratitude for your thorough review and valuable comments and suggestions on our manuscript. We have carefully read your feedback and made corresponding revisions to the paper to address your concerns. Below are our responses and solutions to each of the issues raised in your review.

Comment 1: Lack of clarity in the highlighted abstract section.

Response: Thank you for your valuable comments on our manuscript. We have carefully considered your suggestions and made the following revisions:

We have clarified the description of our data analysis methods, specifically with regard to the use of principal component analysis (PCA) and support vector machine (SVM), and have highlighted these modifications in the revised text.

We have also addressed your concerns about the accuracy of individual subject analysis, and have added a sentence to the Abstract to acknowledge individual differences in the relationship between cerebral blood oxygen levels and motion sickness symptoms.

We hope these changes have addressed your concerns and improved the clarity of our manuscript. Thank you again for your helpful feedback.

Comment2: The citation doesn't support the statement, and the reviewer is not aware of any such effects.

Response: We greatly appreciate your pointing out the misunderstanding in our citation of the literature. Upon your reminder, we have revisited the original article (The Potential Use of Virtual Reality in Vestibular Rehabilitation of Motion Sickness) and its abstract, and indeed found that our previous statement deviated from the original author's intent. To more accurately reflect the original meaning, we have now revised the relevant content in our manuscript to: "severe cases can cause discomfort, including headache, and cold sweats, which may negatively impact one's health". We have also highlighted the revised content in our paper.

Comment 3: The direct causation of motion sickness comes from the conflict between human visual and proprioceptive perceptions of spatial orientation. This is also incorrect; the conflict source is expectation and measurement, not a specific sensory modality. Authors should also use "is thought to" to better reflect the state of uncertainty, as this statement has not been proven!

Response: Thank you for pointing out the inaccuracy in our statement regarding the direct causation of motion sickness. In light of your feedback, we have revised the sentence to better reflect the current understanding and state of uncertainty in this area. The updated sentence now reads: "The direct causation of motion sickness is thought to come from the conflict between human expectations and measurements of spatial orientation, rather than a specific sensory modality." We have also highlighted the revised content in our paper.

Comment 4: A concern was raised regarding reference [5], which was cited to support the general concept of stress response, but did not directly address the relationship between the prefrontal cortex and motion sickness.

Response: Thank you for your comments. We have carefully considered your feedback and have revised the relevant section accordingly. We apologize for any confusion caused by the imprecise language used in the previous version. We would like to highlight that the revised sentence, "The Prefrontal Cortex (PFC) plays an essential role at the level of driving, as indicated by differences in brain activity observed in fNIRS studies," accurately reflects the findings of the article. Additionally, we have made sure to clearly indicate this statement in the revised version of the manuscript.

Comment 5: An issue was raised regarding our citation of a master thesis in reference [12]. Response: Thank you for bringing this to our attention. We appreciate your suggestion to use a more reliable and relevant source, and have accordingly revised our manuscript to cite Irmak, T., Pool, D.M. & Happee, R.'s article "Objective and subjective responses to motion sickness: the group and the individual" in Experimental Brain Research, which provides a more appropriate and up-to-date source for our research. The revised citation is now included in our manuscript as reference [12].

Comment 6: There is an issue of clarity regarding the sentence "Motion sickness occurs due to a close relationship with the person's perceptual nerves."

Response: Thank you for your feedback on our manuscript. We have revised the sentence you mentioned to address the clarity issue. The updated sentence now reads, "Motion sickness is thought to be caused by a conflict between the vestibular, visual, and other proprioceptive systems, which are involved in the perception of motion." We have also added a reference (Koch, A.; Cascorbi, I.; Westhofen, M.; Dafotakis, M.; Klapa, S.; Kuhtz-Buschbeck, J.P. The Neurophysiology and Treatment of Motion Sickness. Dtsch Arztebl Int 2018, doi:10.3238/arztebl.2018.0687.) to support this statement. Additionally, we have highlighted this reference in the text for clarity. Thank you for your helpful comments.

Comment 7: "The statement 'The prefrontal area of the brain is important to human perception' needs to be supported by citations, I couldn't actually find anything from a brief search。"

Response: Thank you very much for your review and valuable feedback on our manuscript. Regarding your concern about the need for supporting citations on the importance of the prefrontal cortex to human perception, we have added a reference (Michel, Matthias. (2022). Conscious Perception and the Prefrontal Cortex A Review. Journal of Consciousness Studies. 29. 115-157. 10.53765/20512201.29.7.115.) as suggested. The abstract of this article discusses the importance of the prefrontal cortex in conscious perception, answering the question of whether prefrontal involvement is directly related to perceptual experience, and introduces relevant correlational and causal evidence. We have added this reference in our manuscript and highlighted it accordingly. Thank you again for your guidance and assistance!

Comment 8: The reviewer has asked for clarification on the statement "fNIRS has advantages such as high spatial and temporal resolution, as well as ease of operation." Specifically, the reviewer has asked how high the spatial and temporal resolution is.

Response: Thank you for raising the question regarding the temporal and spatial resolution of fNIRS. Typically, the time resolution of fNIRS ranges from 100 milliseconds to 1 second, and the spatial resolution ranges from 1 to 3 centimeters. These data indicate that fNIRS can provide relatively precise temporal and spatial analysis of brain activity. In response to your question, we have added this information to our manuscript and highlighted it.

Comment 9: A figure showing the processing steps and their effects on the data is needed to better illustrate the preprocessing process mentioned in the sentence "the architecture of the whole system is shown. This study aimed to develop a classification model for the degree of motion sickness. The entire process consists of 4 main steps. First, the cerebral blood oxygen signal was preprocessed, including mean filtering, median filtering."

Response: Thank you for your comment regarding the processing steps in our study. We appreciate your feedback on the need for a more detailed illustration of the preprocessing steps. As you pointed out, we did briefly mention the preprocessing steps in Figure 3, which included mean filtering, median filtering, and motion artifact suppression (MAS). However, we agree that this may not be clear enough to fully demonstrate the data preprocessing process. Therefore, we have created a new Figure 3 to illustrate each step in more detail, including motion artifact suppression (MAS). The revised figure and its caption have been included in the revised manuscript, and we have highlighted the processing steps in the text.

Additionally, we have added a supplementary figure that provides a more comprehensive overview of the preprocessing steps. The figure and its caption have also been highlighted in the revised manuscript.

Figure 3.1 (a-d). illustrates the fNIRS data preprocessing process, including the original data, Meanfilt data, MedFilt data, and MAS-corrected data.

Comment 10: The statement “Based on her six samples, a total of six first principal component PC1 values and six second principal component PC2 values were derived” needs further clarification. It is not clear what is meant by this statement. The authors should provide a clearer explanation.

Response: Thank you very much for your careful reading of our manuscript and valuable comments. Regarding the content of PCA dimensionality reduction, we have already explained the meaning of PC1 and PC2 in the text, and we have further improved the explanation in the revised manuscript by highlighting the key points in the text. Specifically, PC1 is the direction with the maximum variance of the linear combination of the original variables, while PC2 is the direction with the maximum variance in the direction perpendicular to PC1. We also show in Figure 4 the projection values of each channel on PC1 and PC2, as well as the most representative channels in each segment. If there is still any unclear information, please kindly let us know, and we will be happy to further clarify and improve the manuscript.

Comment 11: The citation used to support the claim that frequency bands have an important role in the study of motion sickness has nothing to do with motion sickness. Please clarify the connection between the cited literature and motion sickness.

Response: Thank you for your comment. After considering your suggestion, we have updated the manuscript to include additional references (Bertolini, G.; Straumann, D. Moving in a Moving World: A Review on Vestibular Motion Sickness. Front Neurol 2016, 7, doi:10.3389/fneur.2016.00014 and Wang, X.; Ma, L.-C.; Shahdadian, S.; Wu, A.; Truong, N.C.D.; Liu, H. Metabolic Connectivity and Hemodynamic-Metabolic Coherence of Human Prefrontal Cortex at Rest and Post Photobiomodulation Assessed by Dual-Channel Broadband NIRS. Metabolites 2022, 12, 42, doi:10.3390/metabo12010042) to support the importance of these frequency bands in the context of motion sickness and cerebral metabolism. We hope that this update adequately addresses your concern.

Comment 12: The reviewer is asking for clarification on the term "infinite-length trigonometric function" used in the context of Short Time Fourier Transform. They would like a more specific definition of this term.

Response: Thank you for your comment. To clarify, in the Short Time Fourier Transform, the basic component of the fixed window function is typically a complex exponential function, which is a periodic function in both time and frequency domains and therefore has infinite length. However, in this study, to achieve time-frequency conversion and enhance the resolution of the signal, we replaced the infinite-length trigonometric function with a finite decayable wavelet basis. We have made this modification in the main text and highlighted it for clarity.

Commit 13: The reviewer suggested that in addition to the 5-fold cross-validation approach, the authors should show sickness and classified sickness as a function of time using line plots for each individual on the validation set.

Response: Thank you for the reviewer's feedback. In order to further validate the effectiveness of our motion sickness classification model, we trained the model separately on each subject's data and generated 13 confusion matrix plots. These plots, which show the true positive, false positive, true negative, and false negative values for each subject's data, provide a detailed understanding of the classification performance of our model. We also created line plots of each subject's motion sickness classification results over time based on their data, reflecting the subjects' motion sickness status during the experiment. These plots, along with our detailed explanations and analyses in the manuscript, demonstrate the effectiveness and robustness of our classification model. We have included these figures in the manuscript to support our findings.

Figure 8-1. Classification of dizziness severity for subject 1.

Figure 8-2. Classification of dizziness severity for subjects 2-5.

Figure 8-3. Classification of dizziness severity for subjects 6-9.

Figure 8-4. Classification of dizziness severity for subjects 10-13.

Commit 14: The manuscript lacks a discussion of findings.

Response: Thank you for your valuable feedback. We have thoroughly revised our manuscript to address the points you raised. Specifically, we have added a discussion section that explores the limitations and potential future improvements of our study. We hope that this section provides readers with a comprehensive understanding of our research and encourages further investigations into this field.

Commit 15: The reviewer raised a concern regarding the contribution of each frequency band to the classification accuracy, as it was not explicitly stated in the manuscript.

Response: We are grateful for the valuable comments provided by the reviewer. Following your suggestion, we conducted a correlation analysis to investigate the contribution of each frequency band to the classification accuracy. Our data showed that there was a certain correlation between the PSE characteristics of frequency bands I, III, and IV and the subjective evaluation of motion sickness, with correlation coefficients of 0.6, 0.2, and 0.19, respectively. We have added these results to the conclusion section of our paper.

Comment 16: Better (also) cite the paper resulting from the same data: Irmak, T., Pool, D.M. & Happee, R. Objective and subjective responses to motion sickness: the group and the individual. Exp Brain Res 239, 515–531 (2021). https://doi.org/10.1007/s00221-020-05986-6

Response: Thank you for your reminder. We have already responded to a similar question (comment 5).

Reviewer 2 Report

Title of the paper: Study of motion sickness model based on fNIRS multiband features during car rides

General observations: The paper investigates the motion sickness phenomenon during car rides using the functional near-infrared spectroscopy (fNIRS) technique during vehicle testing. Further, it examines the relationship between motion sickness and prefrontal cerebral blood oxygen concentration. Principal component analysis (PCA) and Wavelet decomposition were applied to extract the main channel data and the power spectrum entropy (PSE) features of five frequency bands. The support vector machine (SVM) was applied to prepare the motion sickness classification model, with a classification accuracy reached 87.3%. The research also revealed that the magnitude of motion sickness during the ride was closely related to the change in the PSE of the five frequency bands of cerebral prefrontal blood oxygen.

 (a)   There are some typos, for instance, in line no.67 “ease of operation. [17]. This’

(b) The Motion Sickness Severity Scale (MSSS) and the Motion Sickness Assessment Questionnaire (MSAQ), play an important role in accurately capturing data hence authors need to provide more information justifying its validity and accuracy, whether the visuospatial performance test was conducted before the actual ride or not, etc.

(c)   Authors may also include information on the medical history of 13 subjects employed.

(d)   Please refer the line 177: “The values for both PC1 and PC2 vary from -0.8 to 0.8.” may be verified.

(e)   Please refer to “Figure 2. Subjects wearing the fNIRS device and its principles.” It may be suitably modified.

(f)    Similarly, the Figure 5,6, and 8 captions may be modified suitably, and extra text may be shifted to the manuscript accordingly.

(g)   Please refer the line no.192: “According to Nyquist's sampling theorem, the sampling frequency…” reference may be given.

(h)   Several equations are not referred to in the text.

(i)     Authors may provide future scope for research, how this model can be extended to other areas?

(j)     Researchers may also provide assumptions taken during the ride.

(k)   The limitations of the study may also be provided.

(l) Some of the similar studies may also be discussed, for instance:

Zhang, C., Li, S., Li, Y., Li, S.E. and Nie, B., 2020. Analysis of motion sickness associated brain activity using fNIRS: a driving simulator study. IEEE Access8, pp.207415-207425.

Tan, R., Li, W., Hu, F., Xiao, X., Li, S., Xing, Y., Wang, H. and Cao, D., 2022, October. Motion sickness detection for intelligent vehicles: A wearable-device-based approach. In 2022 IEEE 25th International Conference on Intelligent Transportation Systems (ITSC) (pp. 4355-4362). IEEE.

Author Response

Comment (a): There are some typos, for instance, in line no.67 “ease of operation. [17].

Response: Thank you for your feedback. I have made the necessary corrections regarding the punctuation errors near reference [19] and reference [4]. I have also added a space before the reference list based on the formatting example of MDPI's template. Additionally, I have checked and corrected a line break issue in line 284 of the manuscript. I have also modified the paragraph spacing of the image captions, based on MDPI's template.

Comment (b): The Motion Sickness Severity Scale (MSSS) and the Motion Sickness Assessment Questionnaire (MSAQ), play an important role in accurately capturing data hence authors need to provide more information justifying its validity and accuracy, whether the visuospatial performance test was conducted before the actual ride or not, etc.

Response: Thank you for your valuable comments. The Motion Sickness Severity Scale (MSSS) and the Motion Sickness Assessment Questionnaire (MSAQ) are widely used and well-validated instruments for assessing motion sickness symptoms. In our study, the MSSS and MSAQ were both administered immediately after the participants completed the ride. Our data was collected by asking the participants to rate their symptoms every five minutes using the six-point motion sickness scale, which allows for scores with decimal points (e.g. 0.5, 1.5, 2.5, etc.). The specific scale used in our study is as follows:

Table 2. The Six-Point Motion Sickness Scale: Descriptions and Corresponding Scores for Different Degrees of Motion Sickness Severity

Score

Description

0

No motion sickness at all: participants have no sensation of motion sickness and feel normal.

1

Slight motion sickness: participants have slight motion sickness, but can still carry on with normal activities.

2

Some motion sickness: participants experience motion sickness symptoms and may feel some discomfort, but can still carry on with normal activities.

3

Very motion sickness: participants experience moderate motion sickness symptoms such as dizziness and nausea, which affect their ability to carry on with normal activities.

4

Extreme motion sickness: participants experience severe motion sickness symptoms such as severe dizziness and nausea, which prevent them from carrying on with normal activities.

5

Unbearable motion sickness: participants experience extremely severe motion sickness symptoms such as unbearable dizziness, nausea, and vomiting, which require immediate cessation of the activity and medical attention.

We have added this information to the manuscript to further clarify our data collection process.

Thank you for your helpful feedback, which will help us to improve the quality of our manuscript.

Comment (c): Authors may also include information on the medical history of 13 subjects employed.

Response: Thank you very much for your valuable suggestion. When recruiting participants for a study related to car sickness, we will pay special attention to prevent participants with the following medical history from participating: fainting, inner ear problems, dizziness or other balance issues, epilepsy, cardiovascular diseases, pregnancy, drug and alcohol addiction, etc. If participants have these medical histories, they may be more susceptible to car sickness, and participation in the study may also have adverse effects on their physical health. Prior to recruiting participants, we will conduct medical screening to ensure that their physical health is suitable for participation in the study. We have added this information to the manuscript. Thank you again for your valuable comments.

Comment (d): Please refer the line 177: “The values for both PC1 and PC2 vary from -0.8 to 0.8.” may be verified.

Response: Thank you for the meticulous review of our manuscript. We have checked our data and found that the range of PC1 and PC2 values we previously provided was not accurate enough. After precise calculations and verification, we found that the correct range should be between -0.6 and 0.68, rather than the previously provided range of -0.8 to 0.8. We have revised and supplemented this point in the manuscript. We appreciate the reviewer for pointing out this issue and giving us the opportunity to further improve our research results.

Comment (e): Please refer to “Figure 2. Subjects wearing the fNIRS device and its principles.” It may be suitably modified.

Response: Thank you very much for your careful review and valuable suggestions. We have modified and supplemented the detailed description of Figure 2 according to your suggestions. We have revised the description of Figure 2 to read "Figure 2 shows a subject wearing the fNIRS device, and the magnified portion on the right side of Figure 2 displays the measurement principle of the fNIRS device. Specifically, the red section on the left represents the "Light Source" which includes two types of near-infrared light in the 730nm and 850nm wavelength bands, passing through the brain tissue of the subject's forehead and finally being received by the light intensity sensor, or "Detector"." Additionally, we have modified the figure caption to read " fNIRS Device and Measurement Principle with Subject" Thank you for your valuable comments, and we hope that our modifications meet your requirements.

Comment (f): Similarly, the Figure 5, 6, and 8 captions may be modified suitably, and extra text may be shifted to the manuscript accordingly.

Response: Thank you for your comments regarding the titles of Figures 5, 6, and 8. We have revised the titles to be more concise and informative as follows:

Figure 5: Relationship between Wavelet Layers and I-V Band Range

Figure 6: Experiment Flow Chart with fNIRS Signal Recording

Figure 9: Confusion Matrix Plot of SVM Output for All Subjects' Motion Sickness Levels

We have also updated the numbering of these figures in the revised manuscript. Thank you for bringing this to our attention.

Comment (g): Please refer the line no.192: “According to Nyquist's sampling theorem, the sampling frequency…” reference may be given.

Response: Thank you for your suggestion. We have added a reference to support the statement regarding Nyquist's sampling theorem in the manuscript. The reference added is Arie, R.; Brand, A.; Engelberg, S. Compressive Sensing and Sub-Nyquist Sampling. IEEE Instrum Meas Mag 2020, 23, 94–101, doi:10.1109/MIM.2020.9062696.

Comment (h): Several equations are not referred to in the text.

Response: Thank you for bringing to our attention the issue regarding several equations not being referred to in the text. We have carefully checked our manuscript and have now addressed this concern by adding appropriate references to the equations. Specifically, we have added references to the Lambert-Beer law related equations in Equation (1), Fourier transform related equations in Equations (2-4), power spectral entropy related equations and derivations in Equations (5-9), and support vector machine (SVM) related equations in Equations (10-12). We appreciate your feedback and have taken the necessary steps to improve the clarity of our manuscript.

Comment (i): Authors may provide future scope for research, how this model can be extended to other areas?

Response: Thank you for your valuable feedback. We appreciate your suggestion regarding providing future directions for our study. In future research, we plan to explore the applicability of our model in other related fields, such as sports science and cognitive psychology, where fatigue and cognitive workload are also important factors. Additionally, we also plan to investigate the potential of incorporating other physiological signals, such as electroencephalography (EEG) and electromyography (EMG), to further enhance the accuracy of our model. We will add a section on future directions to the Discussion to address these points. Thank you again for your helpful comments.

Comment (j): Researchers may also provide assumptions taken during the ride.

Response: Thank you for your valuable feedback on our paper. We have carefully considered your suggestion to provide the assumptions taken during the ride in our study. We would like to inform you that in order to reduce the influence of external factors on the experimental results, we selected a fixed route and time for the real-world road tests. Moreover, we recorded some assumptions made during the experimental process, such as fixed speed range, driving time, and the driver's experience, which could be provided to readers when necessary. We will add a sentence to the Materials and Methods section to make this information more explicit.

Comment (k): The limitations of the study may also be provided.

Response: Thank you for your valuable comments. In response to your suggestion regarding the limitations of our study, we acknowledge that there are certain limitations that should be addressed. Firstly, our study employed a small sample size, which may have limited the generalizability of our findings. Secondly, data collection was conducted in a specific geographic location and driving environment, which may have introduced potential confounding factors. Additionally, while we used state-of-the-art analysis techniques such as PCA and wavelet decomposition, there may be other advanced methods that could be explored in future research. We have updated our Discussion section to include a more comprehensive discussion of these limitations and potential avenues for future research.

Comment (l): Some of the similar studies may also be discussed, for instance:

Zhang, C., Li, S., Li, Y., Li, S.E. and Nie, B., 2020. Analysis of motion sickness associated brain activity using fNIRS: a driving simulator study. IEEE Access, 8, pp.207415-207425.

Tan, R., Li, W., Hu, F., Xiao, X., Li, S., Xing, Y., Wang, H. and Cao, D., 2022, October. Motion sickness detection for intelligent vehicles: A wearable-device-based approach. In 2022 IEEE 25th International Conference on Intelligent Transportation Systems (ITSC) (pp. 4355-4362). IEEE.

Response: Thank you for your comment. We appreciate your recommendation of related studies on motion sickness detection using physiological signals. As you mentioned, we have already cited two studies by the team mentioned in the first literature you recommended. Additionally, we have added the second literature you recommended to our introduction as it closely relates to our research on motion sickness in driving. Thank you for your helpful suggestions.

Round 2

Reviewer 2 Report

Thank you for your revised version